# Dynamic Privacy-Preserving Recommendations on Academic Graph Data

Erasmo Purificato [1,2,*], Sabine Wehnert [1,2] and Ernesto William De Luca [1,2]

1 Georg Eckert Institute for International Textbook Research Member of the Leibniz Association, Celler Straße 3, 38114 Brunswick, Germany; sabine.wehnert@gei.de (S.W.); deluca@gei.de (E.W.D.L.)
2 Faculty of Computer Science, Otto von Guericke University Magdeburg, Universitätsplatz 2, 39104 Magdeburg, Germany
* Correspondence: erasmo.purificato@ovgu.de

**Abstract:** In the age of digital information, where the internet and social networks, as well as personalised systems, have become an integral part of everyone's life, it is often challenging to be aware of the amount of data produced daily and, unfortunately, of the potential risks caused by the indiscriminate sharing of personal data. Recently, attention to privacy has grown thanks to the introduction of specific regulations such as the European GDPR. In some fields, including recommender systems, this has inevitably led to a decrease in the amount of usable data, and, occasionally, to significant degradation in performance mainly due to information no longer being attributable to specific individuals. In this article, we present a dynamic privacy-preserving approach for recommendations in an academic context. We aim to implement a personalised system capable of protecting personal data while at the same time allowing sensible and meaningful use of the available data. The proposed approach introduces several *pseudonymisation* procedures based on the design goals described by the European Union Agency for Cybersecurity in their guidelines, in order to dynamically transform entities (e.g., persons) and attributes (e.g., authored papers and research interests) in such a way that any user processing the data are not able to identify individuals. We present a case study using data from researchers of the Georg Eckert Institute for International Textbook Research (Brunswick, Germany). Building a knowledge graph and exploiting a Neo4j database for data management, we first generate several *pseudoN-graphs*, being graphs with different rates of pseudonymised persons. Then, we evaluate our approach by leveraging the graph embedding algorithm *node2vec* to produce recommendations through node relatedness. The recommendations provided by the graphs in different privacy-preserving scenarios are compared with those provided by the fully non-pseudonymised graph, considered as the *baseline* of our evaluation. The experimental results show that, despite the structural modifications to the knowledge graph structure due to the de-identification processes, applying the approach proposed in this article allows for preserving significant performance values in terms of precision.

**Keywords:** privacy-preserving recommenders; dynamic recommenders; pseudonymisation; graph embeddings; word embeddings

## 1. Introduction

Personalised systems, social network platforms and search engines can be considered among the most widespread technologies in the last two decades. Their ubiquity and soaring popularity have led (and also are due) to a massive amount of personal data, opinions, professional and individual interests shared by users in several contexts, from e-commerce to academic research. The advent and fast spread of recommender systems have contributed significantly to the growth of interests in retrieving relevant, personalised information in the scientific environment, mainly in terms of experts [1,2] and paper recommendations [3–5]. It is self-evident to point out that, in the current era of big data

and information overload, having such systems can help in navigating the mass of content being created on a daily basis, especially for academics, for whom not being aware of relevant related work, experts or research projects is a common problem.

However, not all that glitters is gold, and the continuous gathering and processing of users' preferences along with their activities are demanding serious considerations about privacy concerns. Despite privacy issues in personalised and recommender systems being studied for a long time [6,7], it is in the last several years that more emphasis has been placed on this question, in all fields where personalised systems are used, as users were never really aware of the problem, especially about what personal data are being used and how securely it is stored [8–10]. A study conducted by the SAS company in July 2018 shows that almost three-fourths of the survey participants are more concerned about their data privacy now than they were in previous years, expressing worries, among other things, about personal information being shared without consent or its inappropriate use. The study can be found at: https://www.sas.com/content/dam/SAS/documents/ marketing-whitepapers-ebooks/sas-whitepapers/en/data-privacy-110027.pdf (last seen on 18 August 2021).

In such a scenario, developing systems and methodologies that guarantee personal data protection and privacy by design is of paramount importance, as well as mandatory, in certain circumstances, such as in Europe with the introduction of the European Union's General Data Protection Regulation (hereafter "GDPR") in 2016, which became enforceable in 2018. See the complete document at: https://eur-lex.europa.eu/legal-content/EN/ TXT/PDF/?uri=CELEX:32016R0679 (last seen on 18 August 2021). One of the primary aims of GDPR is to give individuals (formally called "data subjects") control over their personal data. Within this objective of control, the use of private information shall be lawful if "*the data subject has given consent to the processing of his or her personal data for one or more specific purposes*" (Art. 6(1)(a) GDPR), leading to situations in which a recommender system would have to manage, at the same time, a set of users where, for some of them, personal data can be used and for others not.

Our work aims to retain the performance in a recommender system while allowing complete personal data protection. The presented approach is applied on the domain of the Georg Eckert Institute for International Textbook Research (hereafter "GEI" or "the institute"-http://www.gei.de/en/home.html, accessed on 18 August 2021). The GEI, a member of the Leibniz Association, conducts international, multidisciplinary, and application-oriented research into school textbooks and educational media, centering on approaches drawn from historical and cultural studies.

We exploit data from researchers of the institute and external collaborators, whose profiles are managed by Pure, the Elsevier's Research Information Management System (https://www.elsevier.com/solutions/pure; last seen on 18 August 2021). Along with personal details, profiles contain the users' job information at the GEI, published works, projects in which they are involved and existing relationships with externals (e.g., co-authorship). Performing a preliminary user study, profiles are enriched with research interests, opinions about the confidence in digital tools and preferences on the items (i.e., papers, books, experts, institutions) they want to have as recommendations.

In this article, a dynamic privacy-preserving approach for recommendations in an academic context is presented, where the term "dynamic" refers to the ability of the system to dynamically adapt to changes in privacy, both for new and existing users. We focus on a graph-based recommendation approach, following recent research that demonstrated recommender systems exploiting knowledge graphs to be effective in addressing issues such as new items and data sparsity [11,12]. Changes due to the addition of new pseudonymised users can lead to considerable rework of the knowledge graph structure, as described in the continuation of this paper.

The proposed approach deals with the above-mentioned concepts of *dynamic recommenders* and *privacy preservation*. In order to properly process personal data which may not be accessible because it belongs to external users or to those who have not provided

their consent to its use, we introduce a strategy for *pseudonymisation*, inspired by the design goals for pseudonymisation techniques described by the European Union Agency for Cybersecurity (ENISA) [13] in their guidelines, while keeping essential semantics in the entities and attributes. Pseudonymisation is defined within the GDPR (Art. 4(5)) as "*the processing of personal data in such a manner that the personal data can no longer be attributed to a specific data subject without the use of additional information*", with the latter being "*kept separately*" and "*subject to technical and organisational measures to ensure that the personal data are not attributed to an identified or identifiable natural person*". Regarding this aspect, our goal is to dynamically transform entities and attributes, such as authored papers and research interests, so that any person processing the data cannot identify individuals but can work with the data at hand in a sensible and meaningful manner. Even if it is arguable that publicly available data (e.g., scientific publications) are not classified as personal data, and hence not strictly subject to de-identification, several guidelines for the use of pseudonymisation solutions, like the one published in 2018 by the German Society for Data Protection and Data Security [14], specify that when pseudonymisation is utilised as a technical protective measure, any possible risk of re-identification of an individual must be removed, by decoupling personal information from other data or properly handling those, as in the presented approach.

Using a Neo4j graph database for handling the data, we evaluate the presented approach leveraging a graph embedding method, namely node2vec [15], in order to compute recommendations in different privacy-preserving scenarios (i.e., from total to partial pseudonymisation) and different configurations of the graph embedding algorithm. Neo4j (https://neo4j.com/, accessed on 18 August 2021) is the leading native graph database and graph platform. Among all graph databases, its leading position was confirmed in 2019 by Gartner and Forrester. Comparing the resulting recommendations with those provided by the fully non-pseudonymised graph being the baseline of our evaluation, we show that, despite the structural modifications to the knowledge graph due to pseudonymisation, the proposed approach is able to preserve significant values of performance in terms of precision (i.e., P@5, P@10, and P@20).

In the following, we want to present the structure of our article: in this section, we introduced the challenges and problems related to privacy preservation, concentrating on our use cases, goals and evaluation settings. In Section 2, some of the relevant works published in the last years about privacy preservation in recommender systems, pseudonymisation techniques and dynamic recommenders are presented. Preliminary definitions and notations are set in Section 3. The proposed approach for dynamic privacy-preserving recommendations is presented in Section 4. The evaluation of the case study is described in Section 5. In Section 7, we present conclusions.

## 2. Related Work

In this section, we discuss relevant related work connected to our approach. Due to the constraints mentioned above, the GDPR imposes on data collection for accurate recommendations. We first analyse the state-of-the-art literature on privacy-preserving recommendations (Section 2.1). Hereby, we also include pseudonymisation methods as a specific task to let the user decide to be de-identified (Section 2.2) or to let him/her decide dynamically, in a constantly evolving work environment, to change his/her privacy settings, even for single research items, as a general task for the dynamic recommender systems (Section 2.3).

### 2.1. Privacy-Preserving Recommender Systems

Stating that the more personal data a recommender collects, the more accurate recommendations users can obtain, personalised recommendation services undesirably make users susceptible to privacy violation issues, despite the undeniable benefits. Existing research works on privacy-preserving recommender systems can be categorised in two groups: *cryptography* and *data obfuscation* (or *data perturbation*). Cryptography solutions

leverage encryption algorithms to secure user information; they generally adopt homomorphic encryption in order to hide personal data [16,17]. Homomorphic encryption is a type of encryption, computing over cyphertexts that allow for obtaining after-decryption results equal to those obtained if the exact computation had been performed on the plaintexts [18]. In this context, Aïmeur et al. [19] presented the Alambic system, whose aim is to separate private data between the service provider and a trusted server. Significant contributions showed that a recommender could profile items without learning the rating users provide or which items they have rated [20]. Badsha et al. [21] proposed a similar approach for privacy-preserving recommendations based on the ElGamal public-key encryption algorithm [22]. Data obfuscation solutions leverage the concept of *differencial privacy* and rely on injecting noise on original data in order to reduce the information leakage from the outputs of the system [23,24]. User profile data are transformed to prevent individuals from being identified by exploiting the data itself or the system's output while trying to retain the same or a similar level of accuracy in the results [25]. The main advantage of these techniques is to be scalable since the transformation usually needs to be applied only to the original data, after which the perturbed data can be used directly. However, obfuscation-based methods suffer from accuracy loss due to the addition of randomness to the data [26].

## 2.2. Pseudonymisation Strategies for Personal Data

Pseudonymisation refers to the process of de-identifying a data subject from its personal data by replacing personal identifiers (i.e., informative attributes that can allow the identification, such as name and email address) with the so-called *pseudonyms* [27]. A pseudonym is formally an identifier of a data subject other than one of the subject's real names [28]. The term "pseudonym" comes from the Greek "pseudonumon" meaning "falsely named" (*pseudo* = false; *onuma* = name), a name other than the real name. In order to avoid this wrong connotation "pseudo equals false", pseudonyms are also referred to as *cryptonyms* or just *nyms*. The ideal pseudonymisation technique renders the data in such a way that individuals can not be re-identified, given the modified information about the individual and the context the person is operating in [29], and that no more links between the same identifiers exist in order to avoid susceptibility to re-identification attacks [30]. For many years before the issuance of the GDPR, pseudonymisation has been an issue in biomedical data. In this domain, state-of-the-art approaches exploit deep learning techniques to detect the respective entity types [31,32]. We do not need to detect the entity types in our use case because they are already stored in a graph database with the respective entity attribute. Therefore, we focus on the methods obfuscating the already detected entities and use only the required information for completing the task. Another essential aspect to consider lies in the GDPR's definition of pseudonymisation, which states that pseudonymised data can be lawfully re-identified by using some additional information kept secret and only made available to the so-called *data controller*, namely the person (or group of persons) who determines the purposes and means of personal data processing (Art. 4(7) GDPR). In this direction, Lehmann [30] introduced an effective oblivious pseudonymisation protocol, which maintains links between the same identifiers only in the required local context, thereby retaining control over the entity types which obtain the same pseudonyms (called *chameleon pseudonyms*) for the same values to keep the data utility even after the manipulation. The term "oblivious" refers to the service which neither learns the sensitive information nor the pseudonyms it produces. Eder et al. [33] performed named entity recognition (NER) and subsequent pseudonymisation on a German email corpus, transforming entities by type (e.g., name, location and URLs). Štarchoň and Pikulík [34] describes a different technique called data *blurring*, where the original meaning of the data gets approximated, such that re-identification is not possible, but the meaning may be preserved. This approach can be helpful for recommendation scenarios. In the literature for privacy-preserving context-aware recommender systems, the use of generalisation hierarchies is widespread, having the same data blurring effect [35]. A

disadvantage of this approach stems from the vast variability of values encountered in the research dataset and, therefore, the effort to formulate the hierarchies. Yao and Liu [35] measure the level of privacy of their graph after all generalisations by using entropy.

### 2.3. Dynamic Recommender Systems

For a recommender system, the property of *dynamism* is a combination of various parameters making the system able to capture implicit or explicit changes in either user or system side and accordingly adapt its recommendations [36]. Although dynamic recommender systems (DRS) are not recognised as a separate branch in the field of recommender systems research, dynamic properties have been widely explored and classified in six parameters: *temporal changes*, *context/environment*, *online/real-time processing*, *novelty*, *serendipity*, and *diversity* [36,37]. The first two mentioned properties are the most prominent and studied. Koren [38] focused attention on the effects of temporal dynamics for collaborative filtering. The proposed paradigm can track time changing behaviour throughout the life span of data; the experiments carried out on the Netflix dataset concluded that including temporal changes in recommender systems improves the accuracy of recommendations. In the last decade, with the spread of deep learning architectures, several works have been produced on temporal dynamics and session-based recommenders by exploiting recurrent neural networks [39,40] and convolutional neural networks [41]. Concerning context and dynamic environment properties, their value for recommender systems has been established by many researchers over the years, stating that adding at least certain contextual information helps provide better recommendations [42]. Recommender systems' practitioners commonly consider as contextual information a specific period of the year, the time of the day, the intent of a user to buy a product or his/her location, but recent studies took into account different aspects, such as dynamic environmental changes due to clinical conditions in the current pandemic period [43] and privacy preservation needs [44,45].

## 3. Preliminaries

Before presenting our approach in detail, we will set preliminary definitions and notations and describe the used knowledge graph.

### 3.1. Definitions

The U.S. National Institute of Standards and Technology (NIST) recommends pseudonymisation as the best practice for protecting personal data. The terminology used in this paper is based on NIST's reports on the protection of personally identifiable information [46] and de-identification of personal information [47], as well as on the ENISA's reports on pseudonymisation techniques and best practices under European GDPR [13,27].

*Personal data*  Any information relating to an identified or identifiable natural person (namely a *data subject*).

*Personally identifiable information (PII)*  Any information about an individual maintained by an agency, including any information that can be used to distinguish or trace an individual's identity, such as name, social security number, date and place of birth, mother's maiden name or biometrics records; and any other information that is linked or linkable to an individual, such as medical, educational, financial and employment information.

*Direct identifiers*  called *directly identifying variables* or *direct identifying data* are defined as data that directly identifies a single individual. Some direct identifiers are *unique* (e.g., social security number, credit card number, healthcare identification or employee number); others are considered as *highly identifying* (e.g., name and address in a dataset could not be unique, but they are very likely to be referable to a specific individual). In a privacy preservation process, it is mandatory to remove or obscure any direct identifier.

*Indirect identifiers*  Known as *quasi-identifiers* or sometimes as *indirectly identifying variables* are identifiers that by themselves do not identify a specific individual but can be combined and linked with other information in order to identify a data subject. Examples of indirect

or quasi-identifiers are the zip code or the date of birth: there will be many people who share the same zip code or the exact date of birth, but undoubtedly not many have both equally.

***De-identification*** General term for any process of removing the association between a set of identifying data and the data subject. De-identification is designed to protect individual identity, making it hard or even impossible to learn if the data in a dataset is related to a specific individual while preserving some dataset's utility for other purposes.

***Re-identification*** Process of attempting to discern the identities that have been removed from de-identified data. Since an important goal of de-identification is to prevent unauthorised re-identification, such attempts are often called *re-identification attacks*.

Regarding de-identification, a number of techniques used in our work deserve to be defined to understand the proposed approach properly. The following de-identification technique definitions are based on the "Complete Guide to Data De-Identification" by Privitar, a leading company in development and adoption of privacy engineering technology (https://www.privitar.com/, accessed on 18 August 2021). The guide can be accessed at: https://www.privitar.com/resources/deidentification-guide-ty/ (last seen on 18 August 2021).

***Redaction*** is based on a core principle of privacy: "*if you don't need data, don't use it*". It is nothing more than removing data by completely dropping a column in a database or replacing it with a constant value for every individual.

***Tokenisation*** is the practice to replace the original value with a new generated random value, namely a *token*. Using this approach, the original data format can be preserved, enabling a scenario where pseudonyms can be used in a meaningful manner (e.g., a real email address erasmo@gei.de could be replaced by the token abcde@xyz.de). When the new token is consistently generated, meaning that the same pseudonym always replaces an original value, we call that *consistent tokenisation*, and it is beneficial when the de-identification process needs to be reversible.

***Perturbation*** is masking the original data by the addition of *random noise* to it. For example, all ages may be randomly adjusted with a fixed amount of years, or dates may be shifted by the same number of days.

***Substitution*** makes use of a mapping table to assign specific replacements to original values. Replacements are not randomly chosen, but based on the mentioned table that defines the substitution for each individual's identifier; they could be *one-to-one* or *many-to-one* that is aggregating several values into a single substitute (e.g., different mapping cities to a unique name identifying the region).

***Field-level encryption*** works on a particular field or set of data. It aims to replace the value of an identifier with an encrypted version of that through a specific encryption key (that is secret in most privacy preservation scenarios), used for reversibility and reproducibility purposes. The length of the resulting pseudonym will vary depending on which encryption algorithm one uses.

***Hashing*** is a well-known technique that creates new values using standard algorithms (e.g., SHA256, SHA1 or MD5) built on mathematical functions. Due to the intrinsic reproducibility property of hashing, very often, a random value (called "*salt*") is added to the original data before generating the hash value. However, as stated by Privitar, "*because hashing is vulnerable to attacks which can lead to uncovering original sensitive values, hashing is not a recommended approach to de-identification*" ( Check "Why hashing will not give you complete protection" at: https://www.privitar.com/blog/hashing-is-not-enough/; last seen on 18 August 2021).

***Generalisation*** is the technique of transforming identifiers' values into more general ones, such as replacing a number with a range (e.g., for age, one can generalise 29 by the interval 25–30). This method is commonly used to de-identify quasi-identifiers, and its primary

goal is to reduce the risk of re-identification by creating several individuals sharing the same identifiers' value. The *k-anonymity* model is used to measure whether there are at least *k* records for any given combination of generalised quasi-identifiers (e.g., age in the range 40–45 and zip code 115xx).

For increasing privacy through generalisation, there are several concepts we employ in this work, which are also described in the following:

*Generalisation Hierarchy* is a tree structure with increasing concept abstraction level towards the root. Such a hierarchy can be obtained from existing taxonomies or hand-crafted for a specific domain. The application for generalisation hierarchies in pseudonymisation is to replace rare concepts with their broader term, increasing the occurrence of the surrogate term and decreasing the chances of re-identification. The rare original concepts are removed after substitution.

*Hypernymy* is a relation between two words—the hypernym and the hyponym—where the hypernym is a more general term, encompassing several other hyponyms (e.g., animal is a hypernym for both cat and dog).

*Word Embeddings* are dense, distributed, and fixed-length word vectors, following the notion of distributional semantics [48]. Popularised in their static form by Mikolov et al. [49], they assign a fixed vector to each word in the vocabulary. The more recent contextual word embedding variant obtained from language models such as BERT [50] conditions the embedding vector values on the surrounding words, enabling the distinction of homonyms given their everyday contextual use. Word embeddings are a state-of-the-art text representation method, and using the vectors of pre-trained models from extensive collections has been shown to perform well on semantic tasks.

*3.2. Notations*

Let $G = \{(e_i, r, e_j) \mid e_i, e_j \in E, r \in R, i \neq j\}$ be the **knowledge graph**, where $E$ is the set of *entities* and $R$ the set of *relations* between two entities.

Each **entity** $e \in E$ is represented by a set of different types of attributes $e = \{(a_1^s, ..., a_n^s, a_1^{ns}, ..., a_m^{ns}, l) \mid a_i^s \in S, a_j^{ns} \in NS, l \in L_E\}$, where $S$ is the set of *sensitive attributes*, $NS$ is the set of *non-sensitive attributes*, and $L_E \neq \emptyset$ is the set of *entity labels*. To associate a set or an attribute to a specific entity, we use the subscript notation, such as $l_e$ representing the label value of entity $e$ or $NS_e$ meaning the set of non-sensitive attributes for the same entity $e$. For each label value, we distinguish a different subset of entities:

- $U \subset E$ is the set of *persons* (or *users*) in which each entity is described by $u = \{e \mid e \in E \wedge l_e = \text{"Person"}\}$;
- $W \subset E$ is the set of *research works* (i.e., *publications*) in which each entity is described by $w = \{e \mid e \in E \wedge l_e = \text{"Research output"}\}$;
- $I \subset E$ is the set of *research interests* (also useful as *fields of study*) in which each entity is described by $i = \{e \mid e \in E \wedge l_e = \text{"Research interest"}\}$;
- $P \subset E$ represents the set of *projects* in which each entity is described by $p = \{e \mid e \in E \wedge l_e = \text{"Project"}\}$;
- $O \subset E$ is the set of *organisational units* (i.e., university, research institutions along with their internal departments) in which each entity is described by $o = \{e \mid e \in E \wedge l_e = \text{"Organisational unit"}\}$.

The described subsets of entities are disjoint ($U \cap W \cap I \cap P \cap O = \emptyset$).

A **relation** $r \in R$ is represented by the set $r = \{(a, l) \mid a \in A_R, l \in L_R\}$, where $a \in A_R = \{0, 1\}$ is the attribute required for the pseudonymisation process described in the next sections and $L_R \neq \emptyset$ represents the set of *relation labels*. Each relation connects specific types of entities with each other, creating specific triples in the knowledge graph:

- *Authorship* $= \{(w, r, u) \mid w \in W, u \in U, r \in R \wedge l_r = \text{"writtenBy"}\}$ is the connection between a person and a research output he/she authored;

- $Interest = \{(u,r,i) \mid u \in U, i \in I, r \in R \land l_r = \text{"interestedIn"}\}$ is the connection between a person and a research interest he/she cares about;
- $Membership = \{(u,r,o) \mid u \in U, o \in O, r \in R, \land l_r = \text{"organisedBy"}\}$ represents the connection between a person and an organisational unit he/she belongs to;
- $Participation = \{(p,r,u) \mid p \in P, u \in U, r \in R \land l_r = \text{"participatedBy"}\}$ is the connection between a project and a person who is taking part in;
- $Management = \{(p,r,o) \mid p \in P, o \in O, r \in R \land l_r = \text{"managedBy"}\}$ represents the connection between a project and the organisational unit managing it.

We also define a **pseudoN-graph** (also referred to as *pseudoN* hereinafter) as a knowledge graph where the *N*% of the persons is *pseudonymised*, meaning that those users are de-identified by following the approaches presented in the next section. For example, in the *pseudo10-graph*, the identifiers of 10% of "*Person*" entities need to be pseudonymised along with the removal or adaptation of any potential re-identifiable links. It is worth clarifying that pseudonymising a certain percentage of users does not lead to having the same amount of pseudonymised entities for the other types.

### 3.3. GEI Knowledge Graph

The knowledge graph used in this work is derived chiefly from the Pure Management System instance of the institute. Along with GEI members' profiles, it contains the users' job information, published works, projects in which they are involved and existing relationships with external persons (e.g., co-authorship). Data related to persons, projects, research outputs, organisational units and every connection between entities is retrieved by using the Pure Web Service v5.18 (https://api.research-repository.uwa.edu.au/ws/api/518/api-docs/index.html; last seen on 18 August 2021) and stored in a Neo4j graph database. In this phase, since one of the de-identification techniques described below is about using an automatic algorithm to pseudonymise publication titles in a meaningful and helpful fashion, only research works written in the English language have been considered.

In order to enhance information on user profiles, we performed a preliminary user study among the GEI members having a job position as "*researcher*". For the presented article, the user study results were used to enrich profiles with research interests and preferences on the items (i.e., publications, experts, institutions) they want to have as recommendations. Further information collected and useful for future research is, for example, the consent or refusal to use and process their personal data once the system is effectively deployed.

After getting the entire set of data described, the resulting knowledge graph comprises 364 nodes (i.e., entities), of which 30 are persons, 9 organisational units, 236 research outputs, 22 research interests and 67 projects. Regarding relations, the knowledge graph is constituted by 556 edges, of which 281 are labeled as "authorship", 79 as "interest", 49 as "membership", 80 as "participation" and 67 as "management". Figures 1 and 2 show, respectively, the distribution of entity and relation types within the GEI knowledge graph.

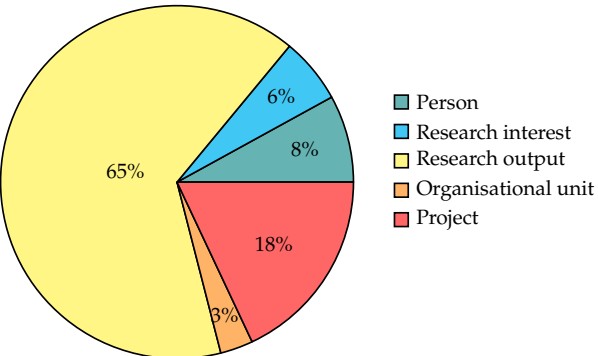

**Figure 1.** Entity types distribution within the knowledge graph.

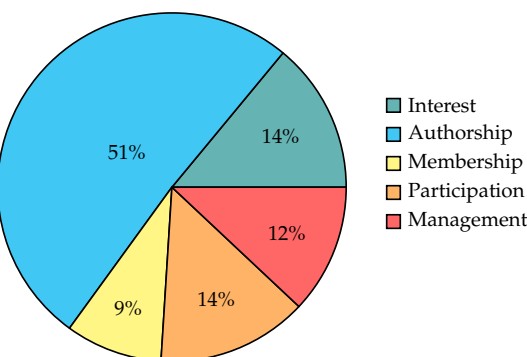

**Figure 2.** Relation types distribution within the knowledge graph.

## 4. Proposed Approach

The core issue addressed by the proposed approach is the *full de-identification* of users to accomplish the goal of guaranteeing privacy preservation while retaining structural information to preserve significant values in terms of precision, namely P@5, P@10 and P@20. In order to achieve the intended objective of de-identification, in addition to the pseudonymisation of the individuals' *personal data* and *personally identifiable information* (PII), the need is to remove or adapt any potential re-identifying links, such as a node with a unique outbound edge towards the pseudonymised user. The presented approach is intended to be applicable in every domain where the need for personal data protection involves publicly available data, such as publications, which must be adequately treated to guarantee complete privacy preservation.

In our application domain, the entities subject to the pseudonymisation and de-identification approach are mainly persons, research outputs and research interests, described, respectively, in Sections 4.2–4.4. The generation of *secrets*, required to control and handle de-identification procedures, is explained in Section 4.1. Algorithm 1 illustrates the pseudocode for the complete procedure.

### 4.1. Pseudonymisation Secrets

As mentioned earlier in this article, one of the crucial aspects of pseudonymisation lies in GDPR (Art. 4(7)). According to the regulation, pseudonymised data can no longer be attributed to a specific individual without the use of additional information that needs to be kept secret and made only available to a group of people (i.e., the data controllers) in charge to determine the aims and means of personal data processing. The adoption of secure additional information, along with suitable de-identification techniques, is mainly required because (1) the pseudonyms should not allow an easy re-identification by any third party (other than data controllers and processors) and (2) it should not be trivial (or even possible) for any third party to reproduce the generated pseudonyms.

In order to accomplish the aforementioned design goals and to ensure a secure pseudonymisation procedure, we follow the approach presented by Lehmann [30] and demand the generation of a **global secret key** to a *central trusted entity* (usually also known as TTP – *trusted third party*), then stored in a *secret trusted storage* only accessible to data controllers. As described in the continuation of this section, the TTP is also responsible for creating secret tokens and the consistent transformation of sensitive attributes in secure pseudonyms. In particular, the secret key is generated using the *ElGamal encryption scheme* [22] which is a well-known encryption scheme that guarantees CPA (*chosen-plaintext attack*) security.

---

**Algorithm 1** Pseudonymisation process

---

1: **procedure** PSEUDONYMISEUSERS(*users_list*)    ▷ *input*: persons to pseudonymise
2:    $sk \leftarrow$ load *global_secret_key* from *secret_trusted_storage*
3:    **for** *user* **in** *users_list* **do**
4:       $uuid \leftarrow get\_uuid(user)$
5:       $token \leftarrow generate\_secret\_token(sk, uuid)$
6:       $pseudo\_uuid \leftarrow generate\_pseudo\_uuid(token, format1)$
7:       store $(uuid, token, pseudo\_uuid)$ in *secret_trusted_storage*
8:       $S_u \leftarrow get\_sensitive\_attributes(user)$
9:       $pseudoS_u \leftarrow pseudonymise\_user\_sensitive\_attributes(S_u)$
10:       $W_u \leftarrow get\_research\_outputs(user)$
11:       $pseudoW_u \leftarrow pseudonymise\_research\_outputs(W_u)$
12:       $replace\_research\_output\_attributes(user, W_u, pseudoW_u)$
13:       $I_u \leftarrow get\_research\_interests(user)$
14:       $generalise\_research\_interests(I_u)$
15:       $replace\_sensitive\_attributes(user, S_u, pseudoS_u)$
16:    **end for**
17:    initialise *processed_users* list
18:    **for** *user* **in** *pseudo_users_list* **do**        ▷ pseudonymised persons
19:       $processed\_users \leftarrow processed\_users.append(user)$
20:       $check\_minimum\_non\_unique\_interests(user, processed\_user)$
21:       $delete\_unique\_research\_interests(user, processed\_users)$
22:       **for** *interest* **in** *deleted_research_interests* **do**
23:          **if** *is_original_research_interest(user, interest)* **then**
24:             $uuid \leftarrow get\_uuid(user)$
25:             $token \leftarrow get\_secret\_token(user)$
26:             $pseudo\_uuid \leftarrow generate\_pseudo\_uuid(token, format2)$
27:             $encrypted\_interest \leftarrow encrypt\_research\_interest(interest)$
28:             store $(pseudo\_uuid, encrypted\_interest)$ in *secret_trusted_storage*
29:          **end if**
30:       **end for**
31:    **end for**
32: **end procedure**

---

To increase robustness and security of our de-identification approach, a *symmetric-key encryption* based on the standard AES algorithm is used to generate a different **secret token** for every entity subject of pseudonymisation, taking in as input both the global secret key and each specific entity's UUID. Symmetric-key encryption (or simply *symmetric encryption*) is a type of encryption where only one key, namely a *secret key*, is used to both encrypt and decrypt information. This process, like all those presented later in the paper, is *consistent*, meaning that different executions produce the same result in order to meet the requirement of *reproducibility* for data controllers. The secret tokens are saved, as the global secret key, in the secret trusted storage mentioned and will be utilised to transform sensitive attributes in unique pseudonyms for each entity in an oblivious manner. No information about the original identifiers will be learned or stored but only replaced as new attribute values. Figure 3 shows the secret key and secret tokens generation process diagram.

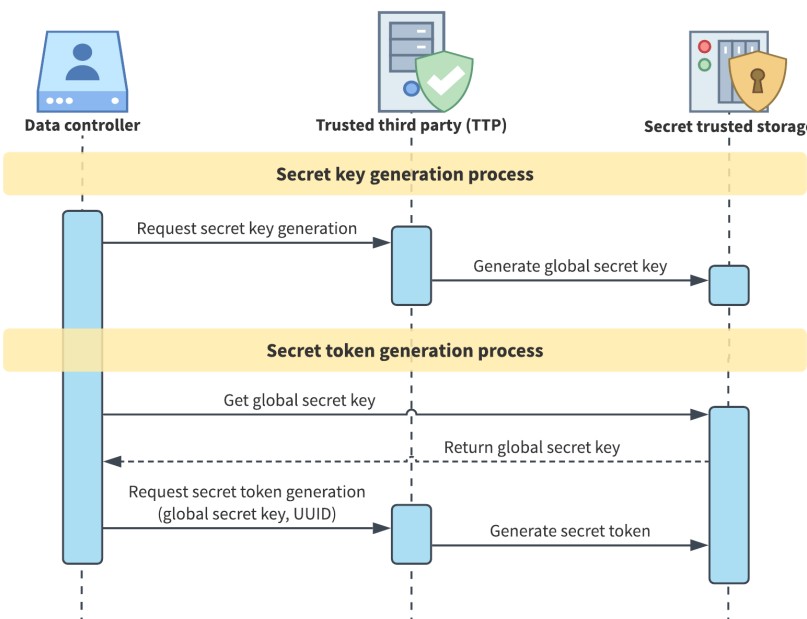

**Figure 3.** Secret key and secret tokens generation.

*4.2. Person Pseudonymisation*

Different de-identification techniques, regarding those defined in Section 3.1, are employed depending on the type of attribute to be pseudonymised. For persons, we identify the set of sensitive attributes $S_U$ = {*UUID, Pure ID, name, email, nationality, employee start date, employee ID*}. Each transformation is performed in such a way as also to achieve the goal of making entities not recognisable as pseudonyms:

- *UUID* is pseudonymised with a two-step procedure, represented in Figure 4: (1) a keyed hash function is applied using the entity secret token as the key and (2) consistent tokenisation produces the pseudo UUID from hashing result. The developed method provides a choice between MD5, SHA1 and SHA256 as hash algorithms. While MD5 and SHA1 are almost comparable, recent research demonstrates that they are less secure than SHA256, but 20–30% faster in being computed [51]. For this reason, and since in the described process it is just the last step of a more complex process than a simple hashing, we employ the MD5 Algorithm by default;
- *Pure ID*, *email* and *employee ID* are pseudonymised via consistent tokenisation using the secret token as the "salt" of the tokenisation function;
- Person's *name* is transformed by substitution. The mapping table is composed of real male and female names and real surnames. According to the actual gender of the user being pseudonymised, a couple (*name, surname*) is picked from the table leveraging the secret token as the *seed* for the consistent choice;
- Finally, generalisation is exploited to de-identify *nationality* and *employee start date*. For the former, countries are generalised with the corresponding continents, whereas, for the latter, the full date is replaced only by the year.

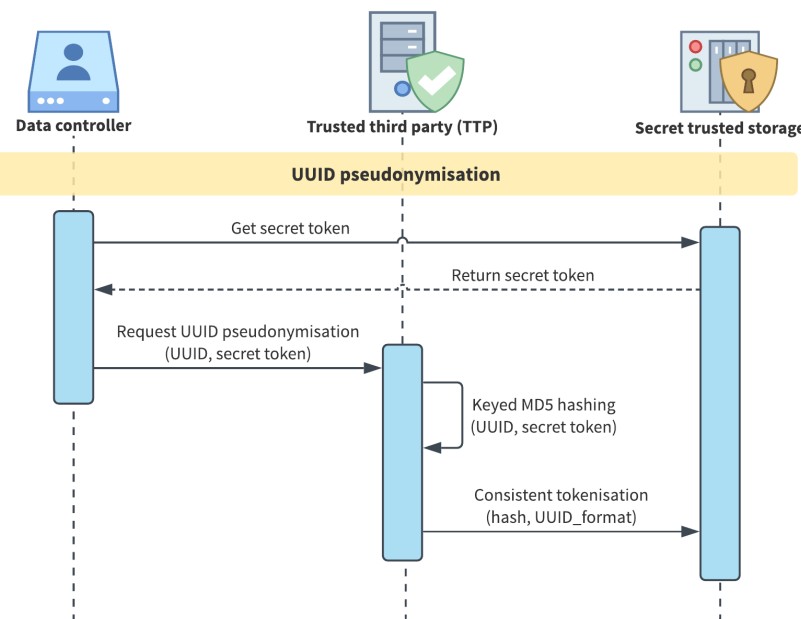

**Figure 4.** UUID pseudonymisation procedure.

*4.3. Research Outputs Pseudonymisation*

For research outputs, the set of sensitive attributes is defined by $S_W$ = {*UUID, Pure ID, title, abstract, number of authors, number of pages, place of publication, ISBN*}. The de-identification techniques applied to the attributes are described below:

- *UUID* is pseudonymised following the same procedure explained for persons in Section 4.2;
- *Pure ID* is transformed by consistent tokenisation using the secret token as the "salt" of the tokenisation function;
- *Title* is an attribute with large variability so that we may not rely on hand-crafted hierarchies in this case. Instead, an unsupervised and scalable method is required to reliably transform the title using higher abstraction levels without losing all of its meaning. Therefore, the title is first reduced to keywords, which are then automatically replaced by their closest hypernym from a set of candidates drawn from the WordNet [52]. To select the keywords, we use a model for dependency parsing provided by the Python spaCy library [53] (https://spacy.io/models/en#en_core_web_lg; last seen on 18 August 2021) and thereby extract nouns, proper nouns and adjectives, which have proven to be useful in preliminary experiments. The best candidate is selected using semantic similarity metrics between a hypernym's word embedding representation and the original keyword or title. We test two methods: first, we compute the Word Mover's Distance [54] using static word embeddings from Google's pre-trained word2vec model (https://code.google.com/archive/p/word2vec/; last seen on 18 August 2021) to compare keyword and hypernym. This method is opposed to the second one of employing the contextual word embedding distance between the hypernym (with optional expansion to synonyms and domains) and the whole text of the title, computed by the BERTScore [55] using the pre-trained SciBERT model [56]. As a baseline, we also compare the results of both methods with the originally extracted keywords instead of choosing their hypernyms. To measure the extent of pseudonymisation gained by each method, we compute the entropy of the resulting keyword distribution across all nodes and select the approach providing the lowest entropy to perform recommendation experiments:

- For the *abstract*, we employ the redaction technique in order to meet the principle as mentioned earlier of privacy "*if you do not need data, do not use it*". The choice is also supported by the fact that this attribute is not present for all research outputs.
- As for a person's nationality, generalisation is used to pseudonymise *place of publication*, replacing city names with the corresponding continent;
- The remaining attributes, i.e., *number of authors*, *number of pages* and *ISBN*, are pseudonymised via consistent tokenisation.

### 4.4. Research Interests Pseudonymisation

About research interests, the pseudonymisation process concerns the modification and the generalisation of the "*interest*" relation with persons. The objective is to remove or adapt any potential re-identifying links, such as an interest with a unique outbound edge towards a pseudonymised user. The approach proposed in the continuation of this section is crucial in our paper, and it is worth noticing that it applies to any knowledge graph where a hierarchical attribute, such as field of study or paper keywords, needs to be de-identified. De-identifying research interests involve the utilisation of a **generalisation hierarchy**. We built a custom hierarchical tree structure for research areas of interest in our application domain derived from the *International Standard Classification of Education* (ISCED-F 2013 – https://tinyurl.com/unesco-org-isced-f-2013-pdf; last seen on 18 August 2021) and the *ACM Computing Classification System* (CCS – https://dl.acm.org/ccs; last seen on 18 August 2021). The hierarchical tree structure is shown after the description of the pseudonymisation process in Figure 5.

To describe the pseudonymisation process, let us start from the simplified and illustrative situation displayed in Figure 6, in which we find three "*Person*" nodes, two of whom share a common interest, and five "*Research interest*" nodes, being "*Knowledge graphs*" (*KG*), "*Recommender systems*" (*RS*), "*Legal artificial intelligence*" (*LAI*), "*Machine learning*" (*ML*) and "*Data science*" (*DS*). The attribute value "0" on the "*interest*" relations indicates that they are the genuine research interests revealed by the users within the user study, as opposed to value "1" representing a pseudo-relation needed for de-identification, as described in a later stage of the procedure.

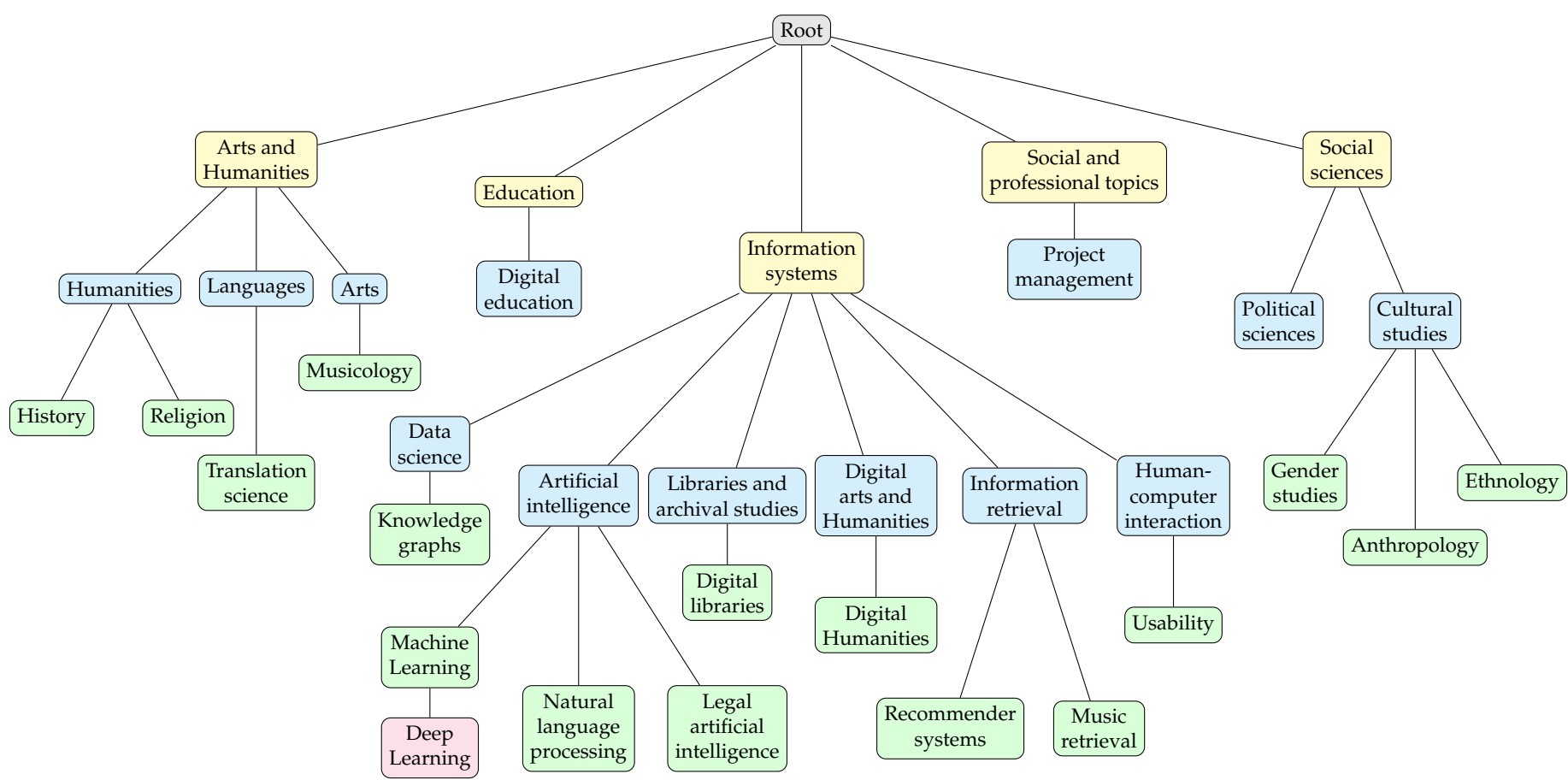

**Figure 5.** Research area tree (different colours refer to different hierarchical levels).

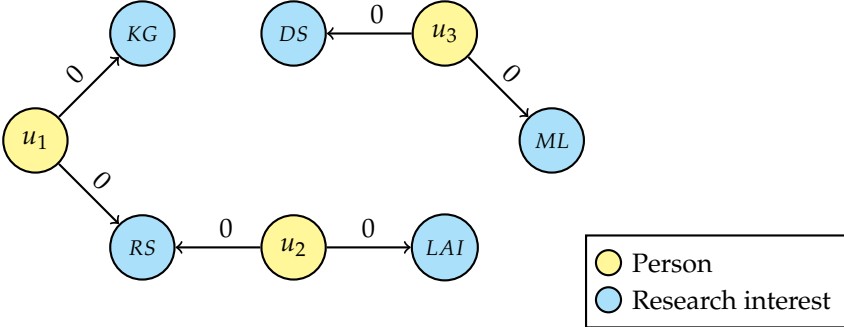

**Figure 6.** Initial graph.

In order to highlight the dynamic nature of the presented method, consider that all three persons from the starting graph should be pseudonymised. The process involves the two main phases with several steps described below.

1.  Processing each person $u_i$ to be pseudonymised sequentially, the first phase of the procedure generalises the research interests that are not connected to other individuals:

    (a) Get the list of unique research interests, namely *KG* for $u_1$, *LAI* for $u_2$, *DS* and *ML* for $u_3$. Note that, depending on the starting person node, some research interests may no longer be unique and therefore are no longer considered for subsequent steps;

    (b) For each person's unique interest, generalise the selected entity by traversing the research area tree structure (Figure 5) from that node up to the root and connecting the user with all nodes in the research area's hierarchy, setting the attribute value to "1" and then breaking the traversal if a link with a non-unique interest is generated. If a research interest is not in the graph, creation of that new node is required before linking, e.g., "*Artificial intelligence*" (*AI*) and "*Information systems*" (*IS*) while generalising *LAI*. Even for this step, the intermediate result depends on the starting person node and two possible scenarios are displayed in Figure 7a (considering the loop sequence $u_1 - u_2 - u_3$) and in Figure 7b (for sequence $u_3 - u_2 - u_1$); it is, however, easy to prove that the final result of the de-identification operation is the same for every loop sequence.

2.  To complete the pseudonymisation process, the second phase aims to check the **minimum-non-unique connection** for each of the original unique research interests of pseudonymised persons $u_i$. It means selecting, among the original and generated "*interest*" relations, those relations which are no longer unique and thus make the user entity not re-identifiable. Processing one pseudonymised person at a time:

    (a) Dealing with person nodes one by one, the starting step is to mark the selected entity as "*processed*". The reason behind this operation will be clear with the next steps;

    (b) For each person's original unique interest (i.e., with attribute value "0"), check whether it is still only connected with the user being processed. If so, the connection will be removed, and the relation is stored in a *secret table* only accessible to data controllers; the correspondent person's and interest's UUID will be saved following the pseudonymisation technique explained in Section 4.2, but using a different tokenisation method for pseudo UUIDs generation in order to meet the principle of "*unlinkability across domains*" [57]. *Unlinkability* ensures that personal data cannot be linked across domains that are constituted by a common purpose and context. It is related to the principles of necessity and data minimisation as well as purpose binding. One of the mechanisms to achieve or support unlinkability is the usage of different identifiers. Saving original interest relations provides the possibility of future restores, in case a

new person with the same research interest will be inserted into the knowledge graph, making that relation valid again;

(c)   Starting from the interest processed in the previous step and browsing the related research hierarchy, find the *lowest* node of the hierarchy not uniquely connected to the pseudonymised person and gradually delete the connections with the traversed nodes (which are necessarily unique);

(d)   Once the *minimum-non-unique connection* has been reached, all connections to any higher nodes in the hierarchy generated earlier must also be deleted. In doing so, it must be borne in mind that the relations may not be unique from this point onwards. If this is the case, in addition to removing the link with the person being processed, it is necessary to check whether this node is connected to only one other user and whether that user has already been processed, since, in that case, the relation must be deleted so that no re-identification is possible.

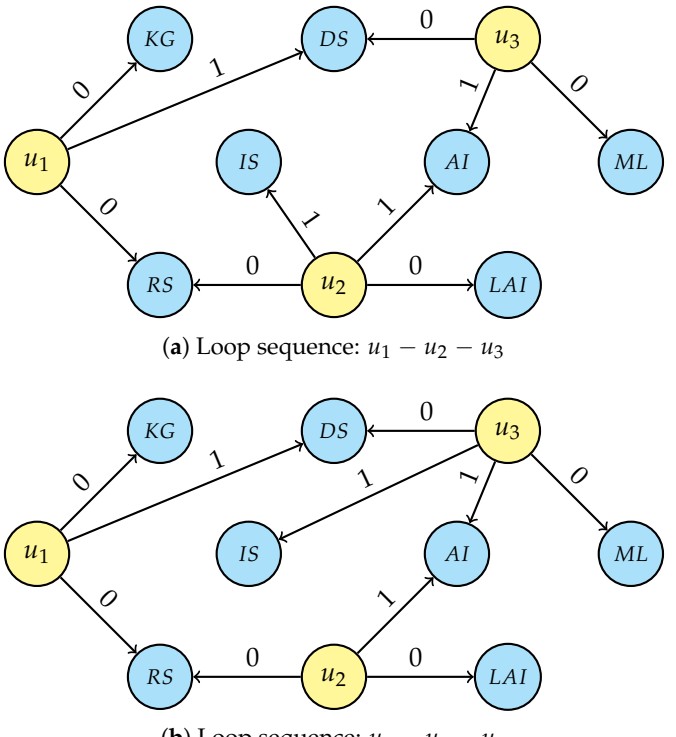

(**a**) Loop sequence: $u_1 - u_2 - u_3$

(**b**) Loop sequence: $u_3 - u_2 - u_1$

**Figure 7.** Generalisation of unique research interests traversing the hierarchical tree.

The resulting graph is shown in Figure 8, and it can be seen that no person can be recognised through any single research interest entity.

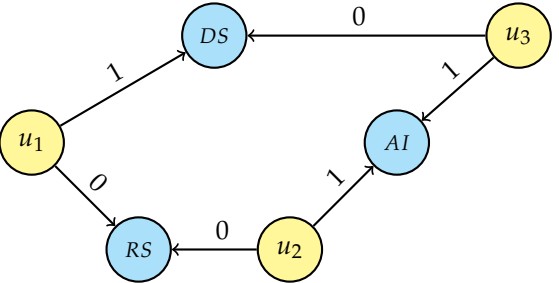

**Figure 8.** Final graph after research interests pseudonymisation.

### 4.5. Adding a New Person

The last part of this section is dedicated to the description of the dynamic approach adopted when a new person $u_n$ is added to the knowledge graph.

Firstly, sensitive attributes in $S_{u_n}$ are pseudonymised according to the procedures presented in Section 4.2. Similarly, for each research output $w_i$ belonging to an "*authorship*" relation with $u_n$, sensitive attributes in $S_{w_i}$ are transformed by the procedures described in Section 4.3.

Regarding user $u_n$'s research interests, let us consider an illustrative scenario where $I_{u_n} = \{KG, DL\}$, being "*Knowledge graphs*" and "*Deep learning*", respectively, as displayed in Figure 9.

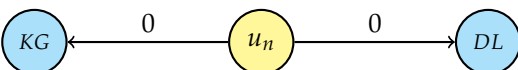

**Figure 9.** Research interests of new person $u_n$.

Following the pseudonymisation method presented in Section 4.4, it is important to dwell on a few points: whenever a new research interest node is inserted into the knowledge graph, it must be checked whether it is one of the nodes removed by a previous de-identification procedure, by querying the *secret table* mentioned in step 2(b), and, if so, proceed to restore the relation and cancel the entry from that table. Figure 10 shows the resulting graph after the first phase of the procedure; the two restored relations can be seen, namely ($u_1$, *interestedIn*, $KG$) and ($u_3$, *interestedIn*, $ML$).

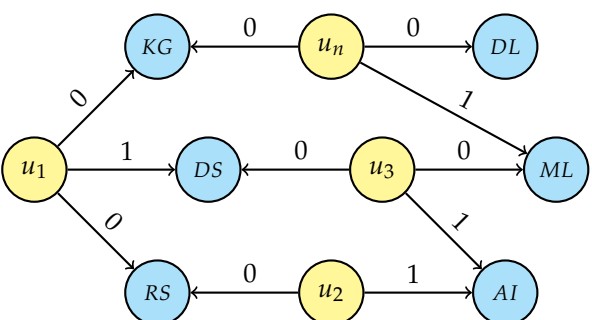

**Figure 10.** Result of research interest pseudonymisation for new person $u_n$ after phase 1.

The final graph, after the execution of the second phase of the pseudonymisation procedure, is displayed in Figure 11. The difference in structure, compared to both Figure 10 and even Figure 8 considering the correspondent persons, is clearly visible and due to the different *minimum-non-unique connections* found during the de-identification process.

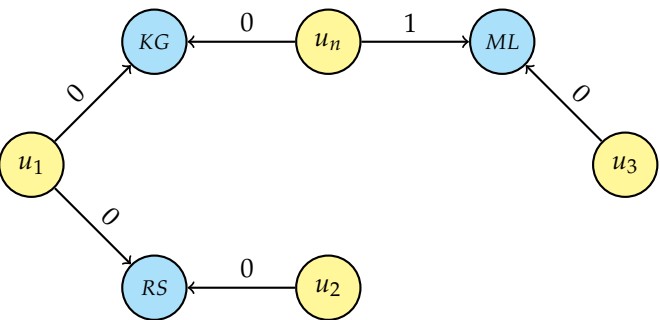

**Figure 11.** Final graph after research interest pseudonymisation for new person $u_n$.

## 5. Evaluation

In this section, the experimental part of our work is presented. The goal is to analyse how the proposed dynamic privacy-preserving approach affects the results of a rec-

ommender system developed within our application domain to evaluate whether the structural modifications due to the pseudonymisation process may lead to acceptable performance preservation.

In our case study, we rely on the graph embeddings algorithm **node2vec** [15] applied on the *GEI knowledge graph* described in Section 3.3. The algorithm can learn representations of nodes in the given graph that are based on their network roles and the neighbourhood they belong to. In recent work, it has also been proven to be effective for item recommendations [58]. Recommendations start from "*Person*" entities following a previous research paper presented by [59] based on the same context, and the knowledge graph is also a continuation of that work. For each person, the items to recommend are chosen based on the results of the preliminary user study we carried out among the GEI researchers.

### 5.1. Experimental Scenarios

To assess the effects of the presented de-identification approach, we implemented a recommender system on the original knowledge graph, leveraging the aforementioned graph embedding algorithm, and then carried out an online evaluation with users belonging to the application domain in order to ascertain the effectiveness of the provided recommendations. Thus, we considered the recommendations from the original knowledge graph as our *baseline*. For the intended evaluation purpose, we generated several *pseudoN-graphs*, with $N \in \{10, 20, 30, 40, 50, 60, 70, 80, 90, 100\}$, and analysed the performance for each of them.

The recommendation strategy and algorithm settings are the same for all scenarios. In particular, we use **node relatedness** as the ranking function to produce item recommendations, likewise [58]. Given $x_u$ as the vector representation of a person $u$ and $x_e$ as the vector representation of an entity $e$, being an item to be recommended to $u$, the relatedness between person and entity vectors is defined as $\rho(u, e) = d(x_u, x_e)$, where $d$ is the *cosine similarity* in our case. Regarding the graph embedding model, most of the hyperparameters are set to their default values as reported in the original node2vec paper [15]. Specifically, *number of walks* is set to 10, *walk length* to 80, *embedding vectors dimension* to 128 and *window size* to 10. For *return parameter p* and *in-out parameter q*, instead, we performed a grid-search delving into the range $[0.25, 0.5, 1, 2]$ for both. These two parameters control how to traverse the graph while computing a random walk and how fast a random walk explores and leaves the neighbourhood of a starting node. Since our approach involves several changes in the graph structure, it is paramount to find the overall best combination to understand which case performance is best preserved.

### 5.2. Experimental Results

Before testing the recommendation performance, we determine the method which yields the most suitable keywords for replacing scientific paper titles. To achieve the highest degree of pseudonymisation while maintaining as much semantic meaning of the title as possible, we employ information extraction methods. Similar to Yao and Liu [35], we measure the extent of privatisation using the entropy, where lower values indicate more privacy. The *entropy* represents the *information value* of a word in a corpus and can be defined as:

$$H(X) = - \sum_{x \in X} p(x) \log p(x)$$

where $p(x)$ is the probability of the word $x$ to occur in corpus $X$.

Table 1 shows the entropy values of the various approaches. The baseline encompasses noun, proper noun and adjective extraction from the title to obtain a reference value for the entropy. Please note that this baseline is considered inappropriate for pseudonymisation, as no transformation of proper nouns is performed at this stage yet. The transformations we consider are using the extracted keywords and their selected hypernyms by static word embeddings or by using the title and compare it to potential hypernyms (additionally with more context added from WordNet synonyms and domains) using contextual word

embeddings from SciBERT. Adding synonyms and domains from the WordNet to the hypernym candidates for context enrichment affects the entropy score negatively and selecting adjectives as keywords in addition to nouns and proper nouns. This indicates the trade-off between the preservation of semantics and the pseudonymisation of the title. The contextual word embeddings achieved the best performance with the lowest entropy with only nouns and proper nouns with an entropy of 5.549 compared to 5.560 for the static word embeddings and 5.934 for the baseline so that all following experiments incorporate title transformations performed by the best performing SciBERT-based pseudonymisation.

**Table 1.** Entropy values for each title replacement strategy (H denotes hypernyms)

| Replacement Strategy | Entropy |
|---|---|
| None (Baseline) | 5.934 |
| Word2Vec (Static; Nouns) | 5.560 |
| SciBERT (Context; H; Nouns) | **5.549** |
| SciBERT (Context: H; Nouns/Adjectives) | 5.591 |
| SciBERT (Context; H + Synonyms + Domains; Nouns/Adjectives) | 5.874 |

The evaluation of our recommender system involves computing precision scores P@5, P@10 and P@20 by comparing the recommendations produced by *pseudoN-graphs* with the baseline for each combination of *p* and *q* parameters. In the recommender systems field, *precision* (P) is defined as:

$$P = \frac{\text{\# of recommended items that are relevant}}{\text{\# of recommended items}}$$

while *precision at k* (P@*k*) is the proportion of recommended items in the top-k set that are relevant (in our case, $k \in [5, 10, 20]$):

$$P@k = \frac{\text{\# of recommended items in the top-k that are relevant}}{\text{\# of recommended items in the top-k}}$$

Excluding *pseudo100* that is the KG where all persons are pseudonymised, the remaining *pseudoN* users to be pseudonymised are chosen randomly. To provide a consistent evaluation, we tested every *pseudoN*, with $N \in \{10, 20, 30, 40, 50, 60, 70, 80, 90\}$, in five pseudo-graph representations, ensuring that the selected pseudonymised persons are different every time, and then computing the average precision values. Evaluation results are extensively displayed in Appendix A (see Figure A1). Considering the baseline recommendations as ideal (i.e., P@5 = P@10 = P@20 = 1), the values in the charts represent the precision measures with respect to every *pseudoN* for each combination of *p* and *q*. All charts have *N* values for *pseudoN* on the *x*-axis and *precision* values on the *y*-axis.

Results show that the best possible combination is $p = 0.5$ and $q = 2$ with average precision measures of P@5 = 0.711, P@10 = 0.645 and P@20 = 0.565. According to the node2vec paper, with $q > 1$ and $p < min(q, 1)$, the generated random walks are biased towards nodes close to the starting node and kept local to its neighbourhood.

## 6. Discussion

In this section, we analyse the results of the performed experiments, discussing the limitations and applicability of the proposed approach in real-world scenarios.

The experimental results displayed in Appendix A (Figure A1), whose best combination has been reported in the previous section, prove that we obtain the best performance when nodes in its neighbourhood represent the person entity. In contrast, we observe the worst cases when the return parameter $p > max(q, 1)$ (in particular, $p = 2$ and $q = 0.5$) that is when the embeddings are built considering entities further away from the starting node, thus not in the close neighbourhood.

Concerning the applicability of our approach in other real-world scenarios, it is helpful in any case where a structured hierarchy is present (e.g., fields of study or topics of interest). The construction of a hierarchical tree, as in Figure 5, remains one of the most critical aspects for applying the proposed approach, which could become even a limitation in case of absence.

In order to further assess the effectiveness of our privacy-preserving approach, we are planning to carry out some experiments on different knowledge graphs with similar characteristics, such as *Microsoft Academic Knowledge Graph* (https://makg.org/; last seen on 18 August 2021), leveraging its "fields of study" entities.

## 7. Conclusions

Our work aims to retain the performance in a recommender system while allowing complete personal data protection. Our work aims to retain the performance in a recommender system while allowing complete personal data protection. In this article, we presented a dynamic approach for privacy-preserving recommendations aiming to retain the performance in an academic recommender system while allowing complete personal data protection, according to the GDPR dispositions. For this purpose, we proposed a de-identification approach based on several European guidelines and state-of-the-art works on pseudonymisation techniques in order to dynamically transform entities and attributes in such a way that any user working on or processing the data will not be able to identify individuals, but can utilise it in a meaningful manner. The presented approach is intended to be applicable in every domain where the need for personal data protection involves publicly available data, such as publications, which must be appropriately treated to guarantee complete privacy preservation. The entire process is handled with the use of *secrets*, accessible only to data controllers that make all procedures consistent and lawful. Procedures include the de-identification of persons and their personal data, research outputs' pseudonymisation and research interests generalisation. Since the proposed de-identification methods lead to changes in the knowledge graph structure depending on the number of pseudonymised persons, evaluation was performed with different scenarios with the purpose to analyse how it would affect the results of a recommender system developed within our application domain, being the Georg Eckert Institute for International Textbook Research. By employing the graph embeddings algorithm called node2vec, applied on the GEI knowledge graph, we compared recommendations produced by pseudo*N*-graphs (where *N* represents the percentage of pseudonymised persons) with those provided by the original knowledge graph, considered as our baseline, measuring values for P@5, P@10 and P@20. Experimental results displayed in Appendix A (Figure A1) show that the dynamic privacy-preserving approach steers towards no-high degradation of performance compared to the original recommendations, suggesting that the obtained outcomes can be considered good, mainly when each entity is represented by an embedding vector that mostly takes into account the node neighbourhood. Future planned research activities for this work involve developing a user interface as the basis for a comprehensive user study to assess the presented approach's effectiveness and measure the usability of the whole system.

**Author Contributions:** Conceptualization, E.P. and S.W.; methodology, E.P. and S.W.; software, E.P. and S.W.; validation, E.P., S.W. and E.W.D.L.; investigation, E.P. and S.W.; writing—original draft preparation, E.P.; writing—review and editing, E.P., S.W. and E.W.D.L.; supervision, E.W.D.L. All authors have read and agreed to the published version of the manuscript.

**Funding:** This research received no external funding.

**Institutional Review Board Statement:** Not applicable.

**Informed Consent Statement:** Not applicable.

**Data Availability Statement:** The data presented in this study are available on request from the corresponding author. The data are not publicly available due to privacy limitations concerning the use of personal information.

**Conflicts of Interest:** The authors declare no conflict of interest.

**Abbreviations**

The following abbreviations are used in this manuscript:

| | |
|---|---|
| CPA | Chosen-plaintext attack |
| ENISA | European Union Agency for Cybersecurity (previously European Network and Information Security Agency) |
| GDPR | General Data Protection Regulation |
| GEI | Georg Eckert Institute for International Textbook Research |
| KG | Knowledge graph |
| NIST | (U.S.) National Institute of Standards and Technology |
| PII | Personally identifiable information |
| TTP | Trusted third party |
| UUID | Universally unique identifier |

**Appendix A. Evaluation Results**

In this appendix, the complete set of the experimental results is shown. As described in Section 5.1, most of the hyperparameters for the graph embedding algorithm (i.e., *number of walks*, *walk lenght*, *embedding vectors dimension* and *window size*) are set to the default values specified in the original node2vec paper [15]. For *return parameter p* and *in-out parameter q*, a grid-search delving into the range $[0.25, 0.5, 1, 2]$ is performed.

Considering the baseline recommendations as ideal (i.e., P@5 = P@10 = P@20 = 1), the values in the charts represent the precision measures with respect to every *pseudoN* for each combination of *p* and *q*. All charts have *N* values for *pseudoN* on the *x*-axis and *precision* values on the *y*-axis.

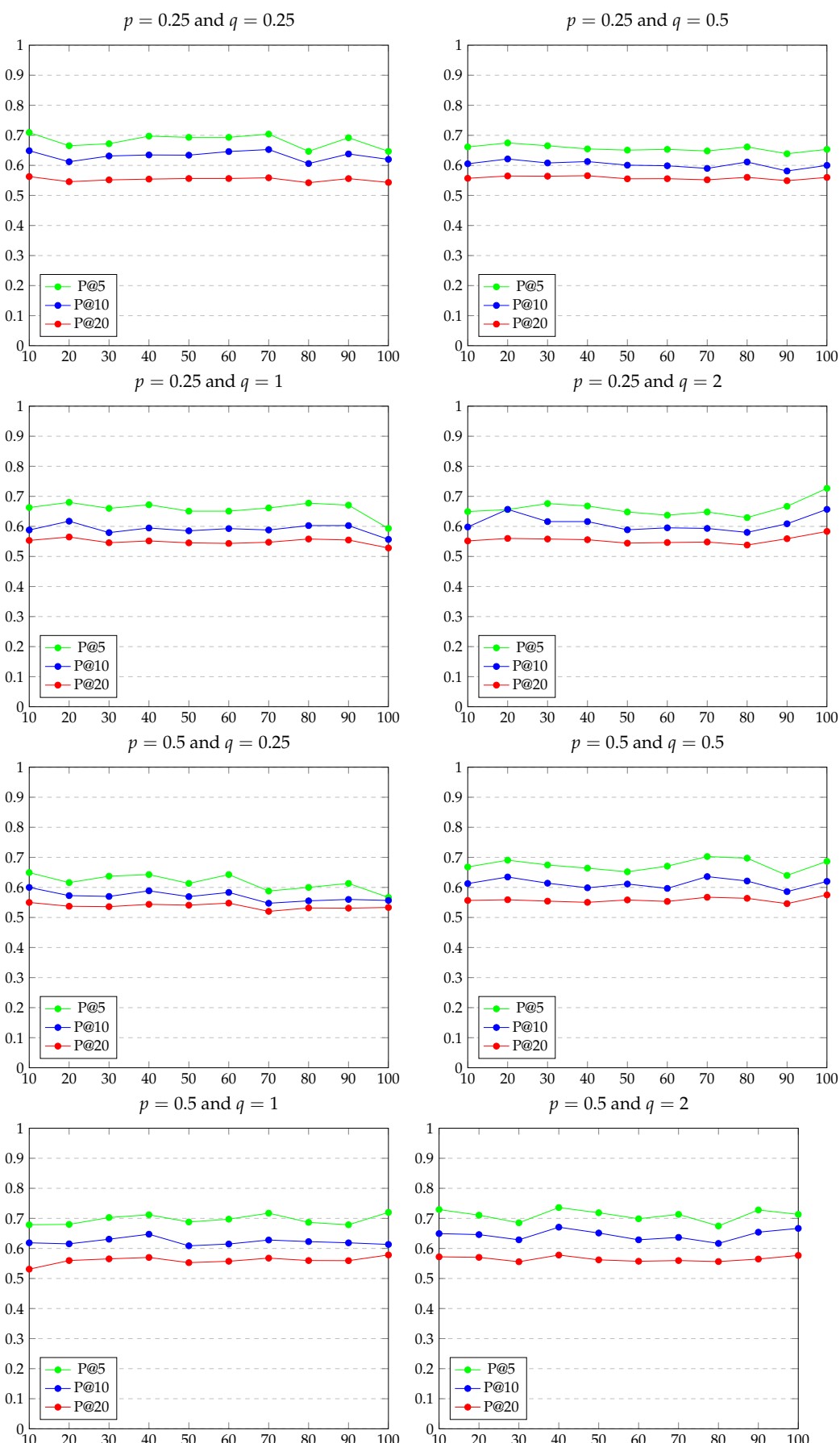

**Figure A1.** *Cont.*

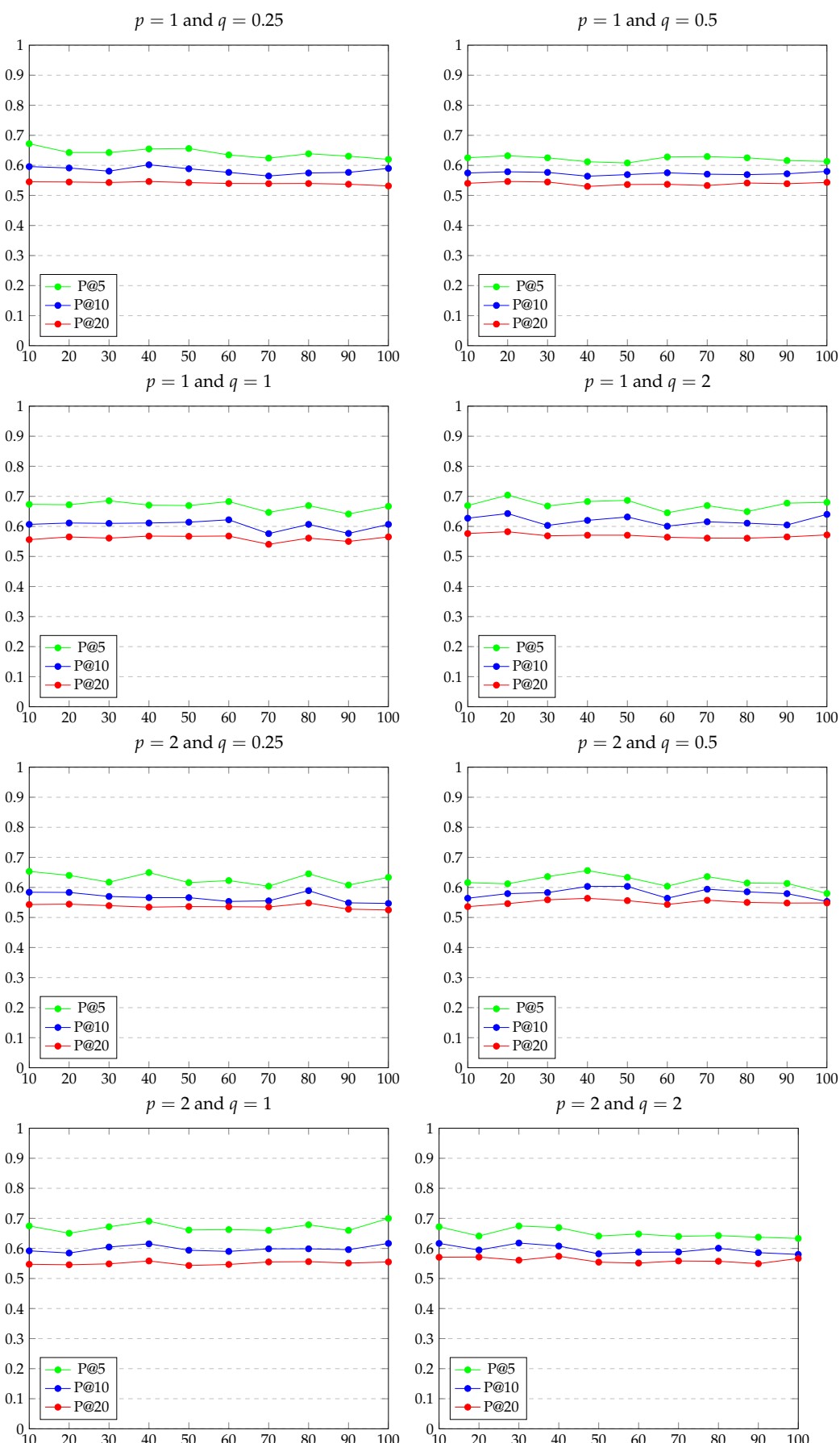

**Figure A1.** Evaluation results

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
