# Peer review of "Dynamic Privacy-Preserving Recommendations on Academic Graph Data"

_computers, doi:10.3390/computers10090107_

Round 1
Reviewer 1 Report
Thank you to the authors for this interesting study about deanonymization.
First of all, I have to remark that I could not see figure 1 and 2 in the manuscript, so they might be missing?
Secondly, I wondered about the motivation of this study to use Neo4J and not Semantic Web technologies like RDF and established vocabularies.
The use of RDF would have alleviated you of defining the graph structure in detail and could have built on previous work in my view.
Could you elaborate on the reasons as to why you would choose neo4j in this study? Is neo4j actually used at your institute for managing that data you research?
Apart from these general questions, I will comment on the individual sections:
The introduction is quite comprehensive and I find it easy to follow, also section 2 gives a good summary about previous approaches with regards to the technologies being applied in the later chapters.
With the preliminaries explained in chapter 3, I also found section 4 to be a suitable approach to assess the performance of your system.
In chapter 5 the experimental setup is explained and in general I agree with the approach. However, could you elaborate on your choice of the ranking function a bit more? Which alternatives would be there and why did you choose node relatedness?
In chapter 6 I am missing an outlook on how your results might impact other knowledge graphs and recommendations systems with different data. Do you think that your results are transferable to these, are there restrictions to be expected and do you have expectations on which other knowledge graphs they would perform well?
Last but not least it would be nice to know if your approach could be applied to a linked data graph in RDF and how such an implementation could be achieved in principle
Reviewer 2 Report
This paper presented a dynamic privacy preserving approach for recommendations in academic context. Overall, the topic is interesting and easy to follow. However, several issues should be addressed to improve the quality of the paper. 1. The motivation of this paper is not clearly presented. I suggest authors rethink the motivation of this research and rewrite the paper. 2. In related work part, authors introduced relevant research from three aspects: privacy-preserving recommender systems, pseudonymisation strategies for personal data and dynamic recommender systems. Why did you conduct literature review from these three aspects? And how these three aspects were related to your research? 3. The theoretical contribution is not clear to me. 4. Figure 1 and 2 is not presented in the paper. 5. Please provide a detailed description of figure 3. 6. The proposed approach is not clearly presented. 7. For evaluation, how did you collect data? How to use proposed approach to make recommendations?Author Response
Please see the attachment.

Reviewer 3 Report
In this article, the authors present a dynamic privacy-preserving approach for recommendations in an academic context. They aim to implement a personalised system capable of protecting personal data while at the same time allowing sensible and meaningful use of the available data.
General Comments:
The title of the paper has the proper length and reflects the main context and contribution of the study.
The references are satisfactory and cover a wide spectrum of the literature on the underlying problem.
It provides sufficient theoretical background to the reader for a research article.
Concerning its structure, it would be better to follow the one suggested by the journal: Introduction, Materials and Methods, Results, Discussion, Conclusion.
The authors have made a thorough description of the proposed approach that combines knowledge graph databases with privacy-preserving. In fact, they provide the pseudocode of the suggested method that summarize the necessary processing steps.
Nonetheless, the extensive sentences and paragraphs render the text hard to follow and maybe incomprehensible.
Some points of major concern are the following:
The abstract needs to be rewritten more focused and without unnecessary details.
The Introduction section needs some enhancement. In particular, in this section, the authors should add more scientific references to support their arguments around the elaborated topic.
A discussion section, before conclusions, should be included to analyse the limitations and the potential issues of this research, emphasizing the differences with previous studies. Hence, the related works section could be omitted and incorporated into the Discussion.
Also, in the Discussion part, the authors should mention the applicability of the elaborated approach in real-world scenarios.
The experiments' environment should be mentioned.
The performance metrics, precision scores and entropy, should be defined mathematically.
Although the authors give details about the experiment's setup, these focused on the best performing results. In which conditions the proposed approach fails to achieve the desired goal? What is the proper range of the involved parameters?
I believe that the article lacks the methodology documentation and the presentation of the simulation results. For the latter, I would recommend the authors emphasize more on the experimentation with various parameters (e.g., number of walks, walk length, window size) to reveal more benefits and different aspects of their method.
Finally, a table of acronyms definitions should be added in the Appendix.
Round 2
Reviewer 2 Report
I appreciate the efforts that authors have made to revise the paper. I accept the revised paper.Reviewer 3 Report
I have no additional remarks on the revised version.